# Toolbox Accelerating Glycomics (TAG): Improving Large-Scale Serum Glycomics and Refinement to Identify SALSA-Modified and Rare Glycans

**DOI:** 10.3390/ijms232113097

**Published:** 2022-10-28

**Authors:** Nobuaki Miura, Hisatoshi Hanamatsu, Ikuko Yokota, Keiko Akasaka-Manya, Hiroshi Manya, Tamao Endo, Yasuro Shinohara, Jun-ichi Furukawa

**Affiliations:** 1Division of Bioinformatics, Niigata University Graduate School of Medical and Dental Sciences, 2-5274 Gakkocho-dori, Niigata 951-8514, Japan; 2Department of Orthopedic Surgery, Hokkaido University Graduate School of Medicine, Kita 21, Nishi 11, Sapporo 001-0021, Japan; 3Division of Glyco-Systems Biology, Institute for Glyco-Core Research, Tokai National Higher Education and Research System, 65 Tsurumai-cho, Nagoya 466-8550, Japan; 4Molecular Glycobiology, Research Team for Mechanism of Aging, Tokyo Metropolitan Geriatric Hospital and Institute of Gerontology, 35-2 Sakaecho, Tokyo 173-0015, Japan; 5Graduate School of Pharmaceutical Sciences, Kinjo Gakuin University, Nagoya 463-8521, Japan

**Keywords:** glycomics, MALDI-TOF-MS, serum, semiautomatic analysis, calibration curve, large-scale analysis, cohort

## Abstract

Glycans are involved in many fundamental cellular processes such as growth, differentiation, and morphogenesis. However, their broad structural diversity makes analysis difficult. Glycomics via mass spectrometry has focused on the composition of glycans, but informatics analysis has not kept pace with the development of instrumentation and measurement techniques. We developed Toolbox Accelerating Glycomics (TAG), in which glycans can be added manually to the glycan list that can be freely designed with labels and sialic acid modifications, and fast processing is possible. In the present work, we improved TAG for large-scale analysis such as cohort analysis of serum samples. The sialic acid linkage-specific alkylamidation (SALSA) method converts differences in linkages such as α2,3- and α2,6-linkages of sialic acids into differences in mass. Glycans modified by SALSA and several structures discovered in recent years were added to the glycan list. A routine to generate calibration curves has been implemented to explore quantitation. These improvements are based on redefinitions of residues and glycans in the TAG List to incorporate information on glycans that could not be attributed because it was not assumed in the previous version of TAG. These functions were verified through analysis of purchased sera and 74 spectra with linearity at the level of *R*^2^ > 0.8 with 81 estimated glycan structures obtained including some candidate of rare glycans such as those with the *N*,*N*’-diacetyllactosediamine structure, suggesting they can be applied to large-scale analyses.

## 1. Introduction

Glycans decorated on proteins and lipids on the cell surface play an important role in homeostasis through their complex interactions and cooperative functions [1,2]. Glycosylation is one of the most common post-translational modifications of proteins, and more than 50% of all mammalian proteins are modified by glycans with large heterogeneity and diversity [3,4]. Therefore, glycomics, which examines the temporal, spatial, and condition-dependent expression states of glycans, has long been a major area of research. In post-genome studies, the transcriptome and proteome have powerful tools available for synthesis and analysis [5,6,7,8,9], but glycans are trace components in biological systems and, unlike nucleic acids, cannot be amplified, making their analysis difficult.

Bioinformatics approaches are important in mass spectrometry (MS)-based glycomics, especially database searches [10,11]. Software has been developed by some groups to determine candidate structures of glycans from MS data using tandem MS (MS/MS) [12,13,14,15,16]. While these software packages allow detailed structure determination using MS/MS, they are not suitable for processing a large number of experiments at high speed. For liquid chromatography (LC)-MS, GlycReSoft [17], which utilizes GlyTouCan [18] for database construction, can be used for the precursor-based analysis. In the case of glycomics for large-scale analysis of serum, software to rapidly identify candidate structures using MS (precursor ion) signals is urgently needed. In addition, to understand the functions of glycans in biological events such as diseases, differentiation, and development, our group developed the concept of the total glycome [19], which is important for observing expression variations linked to various subglycomes such as *N*-glycans, *O*-glycans, glycosphingolipid (GSL) glycans, and free oligosaccharides (FOSs) in response to biological events. To support glycomics with the total glycome in mind, it is essential to design flexible software that can handle various subglycome analyses, such as *O*-glycans, GSL-glycans, and FOSs, with less effort, and treat a variety of glycan labels and sialic acid modifications under different conditions appropriate for each subglycomics from the viewpoint of unified data analysis. Therefore, we have developed Toolbox Accelerating Glycomics (TAG), which allows flexible modification of glycan lists and biosynthetic pathways [20].

TAG consists of several programs, including a ‘TAG List’ that systematically generates the necessary glycan list for *N*-glycans, ‘TAG Expression’ that determines the identified glycans and their abundances, calculates average values, and compares series, *p*-values, etc., from the glycan list and matrix-assisted laser desorption ionization time-of-flight-mass spectrometry (MALDI-TOF-MS) analysis results (Excel file exported from MS and converted to comma separated value (CSV) files), and a ‘TAG pathway’ that maps the obtained expression variation onto the biosynthetic pathway of *N*-glycans and overviews it. Although TAG List is limited to *N*-glycans, it systematically generates glycans lists that are classified into 43 types (for example oligomannose glycans: HM and biantennary neutral glycans: 2_n) according to their known and predicted biosynthetic pathways [20]. The most important feature of TAG is that input files are text files, and output files are Excel-readable CSV files and web browser-readable html files. Thus, input and output files can be shared among various platforms. Furthermore, glycan structures can be easily added to the glycan list because it is a simple text file (CSV). Regarding the speed of execution, analyses that used to take weeks in some cases, even by a skilled operator using Excel macros, can be completed in the order of minutes.

In recent years, improvements in the accuracy of MS instruments and the development of chemical modifications during measurement have made it possible to measure even trace amounts of glycans with high sensitivity [10]. Signals detected in serum that could be attributed to glycans were previously excluded from the TAG glycan list because they were unexpected, such as mannose-rich hybrid structures and *N*,*N*’-diacetyllactosediamine (LacDiNAc) structures [21]. In addition, unusually acidic residues such as glucuronic acid (GlcA) are also observed in *N*-glycans [22]. In our recent development, the chemical modifications that translate α2,3-, α2,6-, and α2,8-sialyl linkages to mass differences facilitated the development of a modified detection method based on sialic acid linkage-specific alkylamidation (SALSA) [23,24,25]. The key to *N*-glycomics of serum is to obtain a list of glycans that can keep pace with these new discoveries and new technologies. It remains difficult to quantitate all glycans and determine their structures since glycan profiles differ among species, tissues, and cells [26].

In the present study, improvements and additions were made to the TAG List and TAG Expression features to address these issues. For sialic acid modifications, the SALSA method is also supported. In addition, GlcA was also introduced into the glycan list. Mannose-rich hybrid structures and LacDiNAc structures mentioned above are also generated in the glycan list by changing the limit on the number of glycan residues. Alongside the improvements in the TAG List, we have also reorganized the internal residues of TAG Expression and generalized the residue information so that it can be applied to GSLs and *O*-glycans without modification. For glycan labels of glycans, the mass can be entered as a parameter. In the previous version of TAG [20], the above-mentioned glycan structures discovered in recent years, which were not assumed not to be expressed because their composition was different from that of typical *N*-glycans, were not included in the TAG List, and carboxylic acids of the sialic acids were methylesterified [19]. Another problem was the large amount of time and effort required to create input files for large-scale analysis. A process for the automatic generation of TAG Expression input files has also been implemented to contribute to the mass analysis of cohorts and other large-scale serum samples. These new functions were verified by analyzing *N*-glycans using purchased sera.

## 2. Results and Discussion

### 2.1. Overview of the Updated Version of TAG

#### 2.1.1. Glycan Residues Handled in the Updated Version of TAG

Table 1 shows the glycan residues used in this study. The complete list of the residues in the current version of TAG is shown in Appendix A. In SALSA, the molecular weights of sialic acids with α2,3- or α2,8-linkages differ from those with α2,6-linkages. This leads to different residues of sialic acids with different linkages. We introduced 3-8NeuAc and 3-8NeuGc for *N*-acetyl neuraminic acid and *N*-glycolyl neuraminic acid with α2,3- or α2,8-linkages, and 6NeuAc and 6NeuGc for those with α2,6-linkages. NeuGc residues are the major components of *N*-glycans in mouse serum. For this reason, we tried to analyze the *N*-glycans of mouse serum using TAG expression including 3-8NeuGc and 6NeuGc, and glycans including those that were detected (data not shown). In addition, triantennary *N*-glycans containing three NeuGc residues were classified into four groups of α2,3- and α2,6-linked sialic acid isomers as expected. These results indicated that the glycan list containing sialic acids with different linkages modified by the SALSA method is effectively working in the TAG list. The residue definitions and SALSA method developed in this study made it possible to assign candidate glycans containing linked isomers of sialic acid. Since the definition of residues shown in Appendix A covers a wide range of residues, the scope of the application can be easily and flexibly extended not only to *N*-glycans but also to other classes of glycans such as GSLs and *O*-glycans.

#### 2.1.2. Modified List of Glycans Containing GlcA, Mannose-Rich Hybrids, and LacDiNAc That Were implemented in the TAG List

Glucuronic acids (GlcA) were also observed as a residue of minor acidic *N*-glycans [22]; however, GlcAs were introduced by competition with sialic acid. In the current version of the TAG List, the summation of the number of NeuAc, NeuGc, and GlcA modifications was redefined as up to 4. The number of GlcA was set to 0 or 1. To represent SALSA, the possible combinations of 3-8NeuAc (3-8NeuGc) and 6NeuAc (6NeuGc) for the number of NeuAc (NeuGc), as shown in Figure 1a for A2 glycan, were considered and generated. Again, the TAG list can efficiently generate a variety of glycans by simply changing the type and range of residues.

As shown in Appendix A, in the biosynthesis of hybrid glycans, oligomannose glycans are trimmed to generate M5 glycans. *N*-acetylglucosamine (GlcNAc) is transferred to M5 glycans to initiate the synthesis of hybrid glycans and elongate the complex chains to synthesize the complete hybrid glycans shown in Figure 1b [2]. Therefore, it was assumed that hexose could be up to three without an M3 core in hybrid glycans. Recently, spectra have been observed that cannot be identified without assuming glycans with more than three hexoses (except M3 core). Hybrid glycans were defined to be M5 + mono-antenary glycans in the previous version of the TAG List [20], with a maximum of 3 Hexes, including 2 Hexes (three Hex in the core structure (Figure 1c) removed from M5) and galactose in the complex chain. Hence, mannose-rich hybrids are defined as compositions with more than 3 Hexes except for the hexoses in the M3 core (Figure 1b). To represent mannose-rich hybrid structures, the number of Hexes must be increased and the maximum value of Hex except for the core hexose is changed from 4 to 6.

Recent observations show that the spectra cannot be identified unless *N*-acetylgalactosamine (GalNAc) is transferred instead of Gal as shown in Figure 1d [21,27]. Gal-GlcNAc is a lactosamine, but it is referred to as LacDiNAc, in which Gal is replaced with GalNAc to yield GalNAc-GlcNAc. The TAG list has been refined to allow the generation of (HexNAc)2 (NeuAc)1 + (Man)3 (GlcNAc)2, (Hex)1 (HexNAc)3 (NeuAc)2 + (Man)3 (GlcNAc)2 and (Hex)2 (HexNAc)4 (NeuAc)3 + (Man)3 (GlcNAc)2 compositions as candidates. Since we now assume that there is only one LacDiNAc structure without galactose residue, if at least one of the sialic acids is 6NeuAc when attributed, it is likely to be an α2,6-sialylated LacDiNAc structure [27]. In these compositions, the number of HexNAc may exceed the number of Hex because GalNAc binds to GlcNAc instead of galactose during chain elongation. For this reason, the previous version of the TAG List [20] did not support LacDiNAc structures. Of course, it is easy to add such structures manually to the glycan list, and there have been uses for them in the past. Users also change the glycan list. In the previous version of TAG List [20], the maximum number of HexNAc was up to the sum of the numbers of NeuAc and NeuGc, with a maximum of 4. In the updated version, the maximum number of HexNAc is up to the sum of acidic residues, plus 1, to create the above LacDiNAc structure.

The size of the list varies greatly depending on the maximum number of each residue. Table 1 also summarizes the range of residues that were used to generate the glycan lists in this study. Basically, it covers all glycans up to tetra-antennary *N*-glycans. The label of the nonreducing terminal is defined as a parameter in the script of TAG List, and in this study, *O*-benzylhydroxylamine (BOA) labeled *N*-glycan with sodium adducts was used. Figure 2 shows the beginning of the TAG List script, which is written in AWK script. The variable wrh in the third (comment line) and fourth lines is the mass of the label of the nonreducing terminal, which can be changed according to the label used in the script. The subsequent lines define the name, maximum number, and mass of the residues used.

#### 2.1.3. Glycomic Data Analysis for Annotation and Quantification of Glycans Using the Updated Version of TAG Expression

An overview of glycomic data analysis using the current version of TAG Expression, its dataflow, and related programs are shown in Figure 3. The main menu of TAG is shown in the center of Figure 3, and the updated functions assigned to each button are summarized in Table 2. A multiple-tabbed Excel file containing sets of peak positions (*m*/*z*) and peak areas was exported from the MALDI-TOF MS instrument; in this ‘masslist’, each tab corresponds to separate MS measurements named ‘experiment’, and a ‘series’ is defined as a group of experiments under certain conditions such as healthy control or disease states. The masslist is processed with a ‘plate file’, which is explained in detail in the next subsection, to create multi-CSV files through a function triggered by the uppermost button of the TAG main menu. Each obtained CSV file corresponds to the tab in the processed mass list. These CSV files are the input files for TAG Expression, which assigns the second and third buttons of the TAG main menu. If the tabs in the masslist are small, the necessary data can be input manually into the masslist, as in the previous version [20]. A function ‘Convert an Excel MS file to multi-CSV’ has been implemented to convert files from xlsx to CSV format (lower most of Table 2).

In the updated version of TAG Expression, a compact summary of the analysis, named ‘summary.csv’, is generated to assist in the efficient attribution of glycans to unknown tissues or organs, or even to sera if the accuracy of MALDI measurement permits. The summary file contains, for each spectra, the attributed glycans, the observed mass, the expression level estimated from the area, its mean value, standard deviation, coefficient of variation (CV), and the Uniform Resource Locator (URL) to allow searching of the attributed structure against the glycan structure database Glyconnect [21]. In addition, glycans whose composition differs by only one residue (e.g., one more Hex or one more HexNAc) from the estimated glycans named ‘linked glycans’ are summarized. This information can be used to examine the validity of the identification of glycans, since glycans before and after based on their biosynthetic pathways’ components, such as precursors, can be known. Figure 4 shows a screenshot of the summary file. MS/MS and other analyses are actually required to confirm whether the structure is actually present, but the summary can be used to make informed predictions.

In TAG Expression analysis, there are two ways to process CSV files: one is analysis to create calibration curves for samples with different concentrations (serum in the current example), and the other is the actual comparison of certain categories as in the previous version. In both analyses, the output files, named ‘exp_list.csv’ and ‘exp_list_zero_cut.csv’, include the structures and expression levels of the attributed glycans, as well as the mean values and significance tests between series. In the former analysis, calibration curves, a scatter diagram of the estimated glycans with error bars and a least-squares line with serum quantities (μL) on the horizontal axis and the expression levels (pmol) on the vertical axis, are output as ‘each_glycan_quant_point.html’. In addition, ‘each_glycan_quant_point_rcut.html’, which summarizes only those with an *R*^2^ value of 0.8 or higher, is also output. In the latter analysis, comparison of among categories, a bar chart with error bars for abundances of each glycan is generated as ‘each_glycan_quant.html’ as in the previous version [20]. The other output files of the analysis are the same as those of the previous version [20] and are summarized in Table 3.

#### 2.1.4. Automatic Generation of Input Files for TAG Expression with Plate Files

When glycomics is applied to cohort studies, it is necessary to perform a considerable number of analyses, in the order of thousands. One of the major challenges is analysis after MS measurement. When performing analyses from thousands of masslists, previous versions of TAG [20] required the user to manually enter information, such as the number of internal standards, tolerances, and categories of measurements. This is a tedious and error-prone process. Although human intervention is necessary at some point, it is prudent to keep manual input as brief as possible to minimize mistakes. To solve this issue, in the updated version of TAG, an Excel sheet plate file is prepared that imitates a MALDI plate. On its first line, common information such as quantities of internal standards, tolerance, and protein levels is placed. After the second line, cells imitating a MALDI plate are prepared, and the type of sample for each spot is included. The series ID (integer number); quantities connected by the ‘_’ (underscore) symbol for making a calibration curve; or series ID and labels of series connected by a ‘#’ (sharp) character for comparing analyses of some categories. In addition, for the analysis of the categories, the label serum1, the label serum2, and the label serum3 were spotted to B22, B23, and B24, as series 1, respectively. Calibrations and analysis of comparisons among categories are processed in separate programs, so there is no problem even if some of the series numbers are the same. This single file can create input files with up to 384 tabs in the masslist. As a result, the probability of making a mistake is reduced, and even if a mistake is made, it is possible to limit the places to check.

### 2.2. Serum Glycomics Using TAG Expression

#### 2.2.1. Generation of Calibration Curves

We present and discuss the results of the calibration curves for 20 serum samples (four samples each of 2 μL, 4 μL, 6 μL, 8 μL, and 10 μL) and the glycans identified, including such as mannose-rich hybrid glycans and LacDiNAc structures. Glycans with NeuGc were removed from the glycan list because NeuGc is absent from normal human serum [28]. Appendix A shows that there are 1686 glycan structures in this list, as shown in Appendix A. The use of calibration curves provides powerful support for quantitative analyses in glycomics. Especially in the case of serum analysis, there is an advantage for exploring the quantification of glycans if several concentrations of standard sera are spotted on the same plate to compare the glycan expression between diseased and nondiseased states. Since precursor peaks are used in the analysis, the reliability of the attributed glycans should always be considered. Using the calibration curve, peaks attributed to concentration-dependent glycan composition can be extracted. In the analysis, we first identified glycans whose masses are close to that of one in the glycan list (within the tolerance range). The reliability of the signal attributed by clustering as shown in the previous paper [20] can be evaluated by concentration dependence by using calibration curves.

Figure 5 shows part of each_glycan_quant_point_rcut.html file, and the whole charts for glycans are shown in Appendix A. The calibration curves show the estimated glycans with *R*^2^ values > 0.8 on the least-squares line including the top four glycans with the highest expression, two mannose-rich hybrid glycans, and two LacDiNAc glycans. The number of spectra in which glycans were found in at least one of the 20 samples was 222. Each of these spectra might be attributed to multiple composite candidates of the same mass. The number of candidate glycans was 252. The overall calibration curves in Appendix A show that there were 74 spectra with linearity at the level of *R*^2^ > 0.8. Appendix A shows the calibration curves for the 81 glycans corresponding to 74 spectra. Recent serum-based glycomics allowed us to identify and quantify no more than ~100 glycans [29,30,31,32,33]. Our current analysis estimates reasonable numbers of *N*-glycans compared with these previous studies. In addition, some degree of quantitation may be possible for these glycans. Table 4 estimated glycan compositions of serum with a hyperlink to the Glyconnect database [34] of the 81 glycans used in the analysis. Hyperlinks to Glyconnect are useful when examining possible glycan structures based on glycan composition.

#### 2.2.2. Identification of Plausible *N*-glycan Structures Added to the Updated Version of TAG List

Table 5 shows the candidate structures estimated for mannose-rich hybrid glycans with more than three Hexes and LacDiNAc structures considered in the TAG List. For both types of glycan, we show those that have been introduced to sialic acid. Figure 6 shows a screenshot from the summary file for the glycans with *R^2^* > 0.8 in the calibration curves. The linked glycans in the fourth column from the left in the summary file in Figure 6 indicate the reliability of the composition of the estimated glycans. The entire summary file is shown in Appendix A and the signals with *R*^2^ > 0.8 are summarized in Appendix A. In this analysis, we used a glycan list containing ~1700 glycans and searched 20 experimental samples. The program took ~3 min to execute using Microsoft Surface 7+ (core i7-1165G7/memory 16 Gb).

For mannose-rich hybrids, most of the glycans are rare glycans that are only expressed in ~3 out of 20 samples, but the two glycans (ID90, 94) shown in Figure 6a are expressed in 19 of 20 samples. The calibration curves for these glycans showed *R*^2^ values greater than 0.8 (Figure 5b). In the former glycan, the α2,6-linkage was attributed using the glycan list with SALSA (Appendix A). The linked glycan is missing 6NeuAc (ID41), which is considered to be the composition of the precursor and is connected in biosynthetic pathways. It is interesting to note that the acidic residue of ID94 is GlcA, which has also quantifiability. The linked glycan of ID94 is the one lacking GlcA (ID58), it is also connected as a pathway.

Most of the putative LacDiNAc glycans are rare and expressed in only 2–6 out of 20 samples. The glycans of ID100 and ID126 shown in Figure 6b are expressed in 20 and 12 out of 20 samples, respectively. In addition, the *R*^2^ values of calibration curves are over 0.8 (0.99 and 0.86 (Figure 5c)). For both compositions, the sialic acid has the α2,6-linkage, suggesting that it is LacDiNAc. This indicates one of the advantages of using SALSA. The linked glycan of ID100 is missing 6NeuAc (ID51), missing Hex (ID69), and one in which HexNAc is added (ID127). The linkage glycan of ID126 is missing 6NeuAc (ID83). These linked glycans are again valid as linkages in the biosynthetic pathway. This suggests that tracking the linked glycans for estimating the glycans introduced in the summary file can be a powerful tool in glycomics. In the future, the development of software that outputs networks using this difference in composition could lead to the generation of detailed glycan structure maps that provide a bird’s eye view of glycan networks.

## 3. Materials and Methods

### 3.1. Materials and Reagents

Human serum and tris(2-carboxyethyl)phosphine hydrochloride (TCEP) were purchased from Sigma-Aldrich (St. Louis, MO, USA). Disialyloctasaccharide (A2GN1) and methylamine (MeNH2) were purchased from FUJI FILM Wako Pure Chemical Corporation (Osaka, Japan). C 57BL/6NCr mice were maintained and six-month-old mice were used for glycemic analysis (*n* = 6). All experiments using laboratory animals were approved by the animal experiment committee of the Tokyo Metropolitan Institute of Gerontology and carried out according to its guidelines (Permit Number: 17010, 18,006). Peptide-*N*-Glycosidase F (PNGase F) was purchased from Roche (Mannheim, Germany). *O*-Benzylhydroxylamine was purchased from Tokyo Chemical Industry (Tokyo, Japan). BlotGlyco beads were purchased from Sumitomo Bakelite Co., Ltd. (Tokyo, Japan). MultiScreen Solvinert 0.45 μm low-binding hydrophilic polytetrafluoroethylene plates were purchased from Merck Millipore (Darmstadt, Germany). Aminooxy-functionalized tryptophanylarginine methyl ester (aoWR) was prepared as previously described [35]. Other solvents and reagents were of the highest grade commercially available.

### 3.2. Quantitative Serum N-Glycan Analysis by MALDI-TOF MS

Serum samples were denatured, digested with trypsin, and heat-inactivated. *N*-glycans were then released with PNGase F as previously described [36]. Released *N*-glycans were captured chemoselectively on BlotGlyco beads, followed by sialic acid linkage-specific alkylamidation of sialic acid residues for stabilization and distinguishing linkages via MS analysis. Using the SALSA procedure, α2,6-linked sialic acids were labeled with isopropylamine and α2,3-linked sialic acids are converted to intramolecular lactone forms in the first step of SALSA condensation. Lactones of α2,3-linked sialic acids can be selectively converted to methylamide forms via ring-opening aminolysis. As a result, the SALSA method can distinguish between α2,3- and α2,6-linked sialic acid isomers with different molecular weight by Mass analysis [24]. Finally, amidated glycans were released from beads, simultaneously labeled, and spotted using an ionic liquid matrix (α-Cyano-4-hydroxycinnamic acid diethylamine salt) onto a MALDI target plate. MALDI-TOF MS analysis was performed on an ultrafleXtreme mass spectrometer (Bruker Daltonics, Yokohama, Japan) in positive ion reflectron mode using proprietary matrix composition. Each sample was spotted in quadruplicate, and spectra were obtained in an automated manner using the AutoXecute feature in flexControl software (Bruker Daltonics, Yokohama, Japan).

### 3.3. Automatic Generation of the Input File for TAG Expression Oriented to Large-Scale Cohort Analyses

#### 3.3.1. Plate File

For analysis of which samples are measured under the same conditions, it was necessary to enter information such as internal standards and series in all masslist tabs for the previous version of TAG [20]. This is a tedious and error-prone process for cohort studies of the order of thousands of data. In the current version of TAG, a plate file that imitates a MALDI plate is introduced to manage which spot and what kind of sample is attached to which spot by one Excel file per plate. The input file for TAG Expression is automatically generated based on the tab name in the masslist and the spot information in the plate file. This process corresponds to the solid arrow part in Figure 3.

Figure 7 shows the procedure for the automatic generation of TAG Expression inputs. The measurement is performed using a MALDI plate to which the sample is attached and the masslist is output. On the other hand, the information of the sample attached to the MALDI plate is entered in the plate file. The input file for TAG Expression is automatically generated by processing the masslist and plate file using the “make input file from Masslist and platefile” button in TAG. The generated file will be placed in the folder where the masslist is located, under a folder named after the .xlsx extension from the masslist. The input files are generated in the calibration folder if you want to create a calibration curve, or in the analysis folder if you want to perform an actual comparison analysis.

#### 3.3.2. Generation of Input Files for TAG Expression Analysis

After clicking on the “Make input file from MassList and plate file” button in the TAG menu shown in Figure 3, a file selection screen for selecting a masslist file and a plate file is displayed in order, and the process is executed to create the input files for TAG Expression. In the same folder where the masslist file is located, a folder with a name (basename) excluding the masslist extension is created. In the case of the calibration curve, a folder named ‘calibration_line’ is created, and in the case of analysis, a folder named ‘analysis’ is created, and the input files are stored in these folders. When executing TAG Expression, these folders are selected as the folders containing input files.

### 3.4. MS Data Analysis Using the Updated Version of TAG Expression

Although TAG Expression has been improved internally, its use in analysis remains unchanged from the previous version. TAG Expression [20] is performed using the second (calibration curve) and third (analysis by category) buttons from the menu in Figure 3. After clicking the button on the menu, a dialog box for selecting a list of glycans is opened, and a list file is selected. Glycans containing NeuGc were excluded from the glycan list because human serum was used in the current analysis. Next, TAG Expression is executed when a folder is selected from the dialog box to select the folder containing the input files created from the masslist. When input files are created using the plate files introduced in this study, a folder named ‘calibration_line’ for calibration and a folder named ‘analysis’ for analysis are created in the folder of the base name of the masslist file, and should be selected as appropriate. If an input file is manually generated as in previous versions [20], the folder that contains the input file created should be selected.

## 4. Conclusions

In this study, we improved TAG List and TAG Expression, and added several new functions to enable analysis using recent measurement techniques for cohort studies and large-scale analyses in the near future. These include the addition of recently detected structures to the TAG List and the automatic generation of input files of the TAG Expression for large-scale analysis. For the calibration function, spots of samples for calibration and samples for analysis on the same plate provide powerful support for analysis, including exploration of quantitation. This analysis can now be performed using a very simple procedure. The introduction of the new residues enables us to analyze sialic acids (NeuAc and NeuGc with α2,3-, α2,6-, and α2,8-linkages), and trace acidic residues such as GlcA. Mannose-rich hybrid glycans and LacDiNAc structures that have been observed in recent years were also included. The summary file provides a simple expression profile, and structures with linked glycans suggest a network, which is useful for information for identification. It can be inferred that the list of linked glycans can create a kind of network, which may be similar in some respects to the approach of Robin et al. [37]. These improvements and recent advances in analytical technology allowed us to attribute over 200 spectra, of which, 74 spectra were quantitative with an accuracy of *R*^2^ > 0.8 relative to the calibration curve. It may also be possible to extract the glycan list from the glycan compositions attributed to these spectra and use it to search for glycans that have quantitative properties in other samples of the same species. Although not included in the present work, modifications such as acetylation, sulfation, and phosphorylation were also defined as residues, hence analysis assuming these modifications is also possible. In addition, since we have standardized the residues used, including those that appear in different classes, such as *O*-glycans and GSL glycans, the barrier for dealing with other classes of glycans has been lowered. We believe that the plate file will make it possible to automatically generate input files for the analysis of the results of experiments, thereby considerably reducing the chance of human error in the analysis.

TAG was developed based on simplicity and flexibility to expand the range of applications. Based on this approach, the program itself functions almost as a filter, and the input file and the list of glycans are text files, while the output file is also a text file, which can be read by Excel or a web browser, respectively. Currently, this system is being developed with MALDI in mind. However, since the only inputs required for TAG are the mass (glycan list), *m*/*z,* and area (masslist) of the glycans in mass spectrometry, it is expected that the system can be adapted to other instruments. If there are any characteristics of signal acquisition that are unique to that instrument, they should be reflected in the algorithm for glycan attribution. We hope that future glycomics studies will be made easier using our or similar software, leading to more glycomics software development.

## Figures and Tables

**Figure 1 ijms-23-13097-f001:**
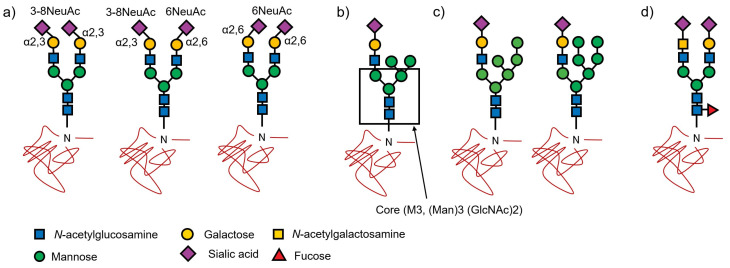
Structures of estimated mannose-rich hybrid glycans and LacDiNAc and a kind of structures of SALSA modification. (**a**) biantennary sialyl glycan (A2). (**b**) One commonly observed hybrid structure. (**c**) Estimated mannose-rich hybrid. (**d**) A LacDiNAc structure.

**Figure 2 ijms-23-13097-f002:**
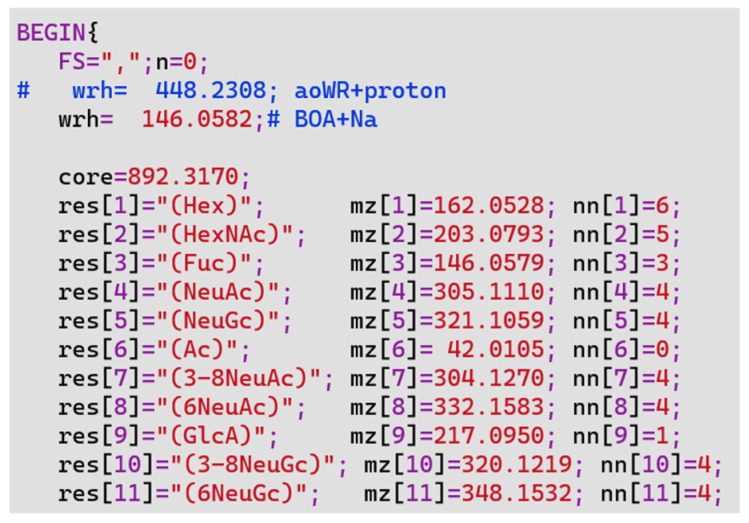
The beginning of the TAG List script. # means that the rest of the line is a comment. Therefore, anything after # in the same line is not executed within the program.

**Figure 3 ijms-23-13097-f003:**
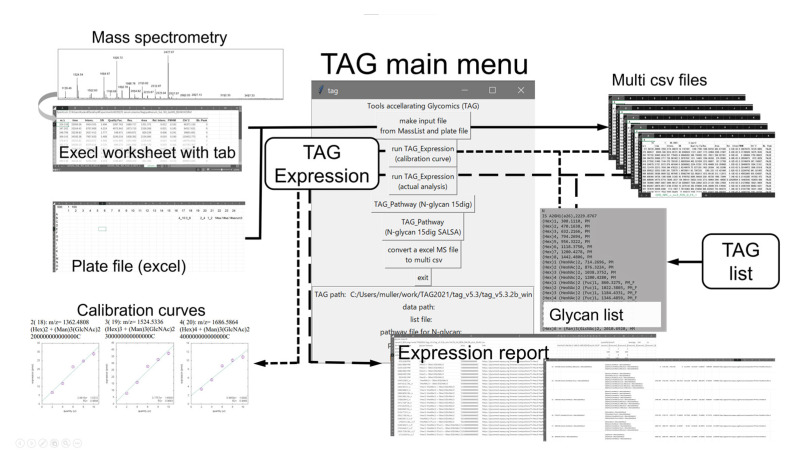
Overview of the updated version of Toolbox Accelerating Glycomics (TAG) showing TAG Expression and related programs.

**Figure 4 ijms-23-13097-f004:**
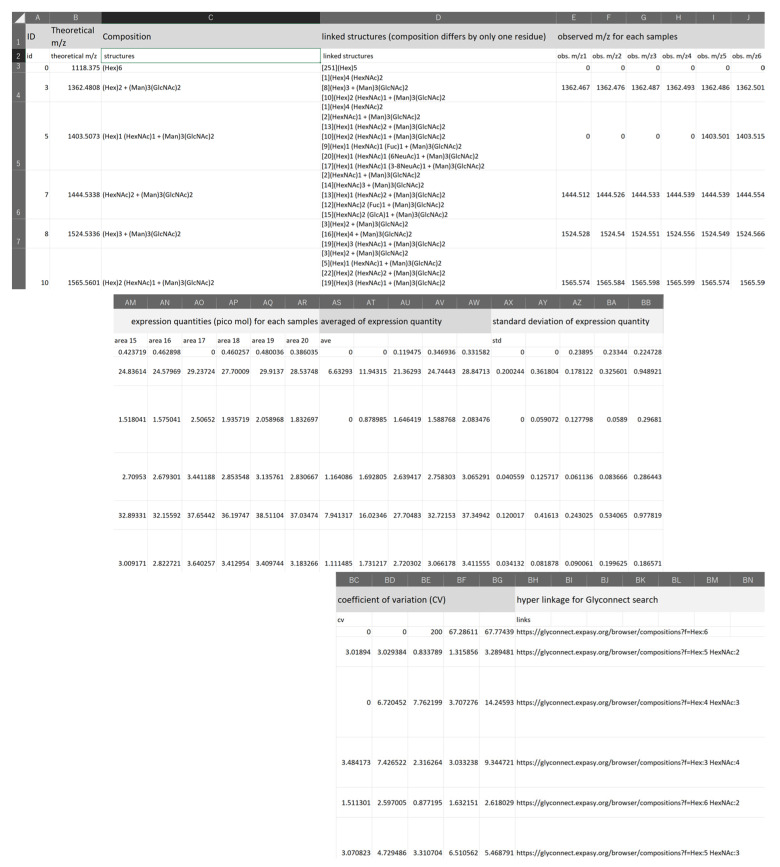
Screenshot of a part of the summary file.

**Figure 5 ijms-23-13097-f005:**
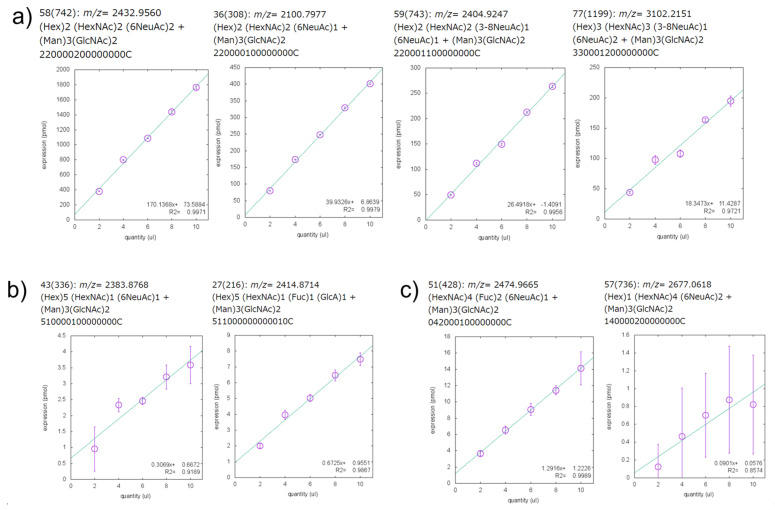
Screenshot of part of each_glycan_quant_point_rcut.html file. (**a**) Calibration curves of the four most abundantly expressed glycans, (**b**) Calibration curves of two mannose-rich hybrid glycans, and (**c**) two glycans with the LacDiNAc structure. Each panel includes *m*/*z*, glycan composition, 15-digit glycan composition ID, and calibration curve graph. The 15-digit glycan composition ID indicates the number of each residue in the order shown in Appendix A. The ‘C’ at the right end indicates the M3 core structure ((Man)3 (GlcNAc)2). In the graph of the calibration curve, the linear function of the completion line and the fitting *R^2^* are shown in the lower right corner.

**Figure 6 ijms-23-13097-f006:**
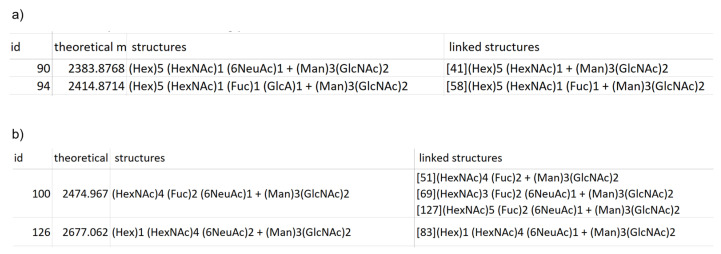
Screenshots of parts with respect to mannose-rich hybrids (**a**) and LacDiNAc (**b**) in the summary file (Appendix A). Each column from left to right shows the following: spectral ID, mass, glycan composition, and the plausible glycan composition that differs by only one residue from the glycan composition on the left. Numbers in square brackets are the spectral IDs.

**Figure 7 ijms-23-13097-f007:**
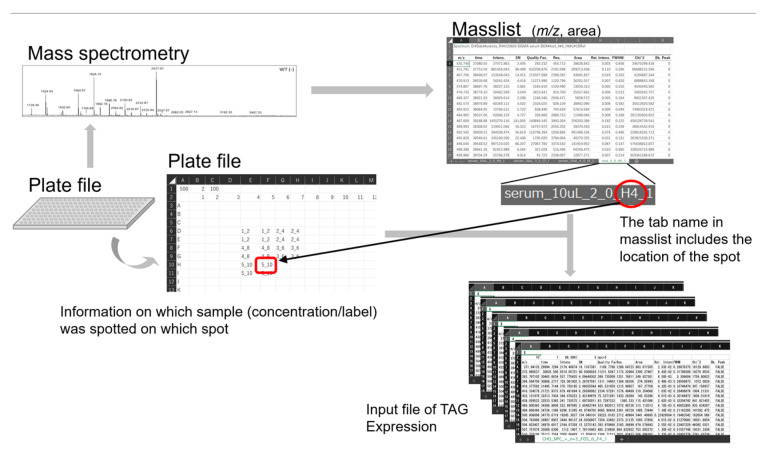
Automatic generation of input files of TAG Expression using plate file.

**Table 1 ijms-23-13097-t001:** Glycan residues in TAG used in this study. The methods used to modify carboxylic acid of sialic acid are shown in parenthesis. ^a^ Residue ranges are shown without the M3 core ((Man)3(GlcNAc)2).

Residue ID	Name	Mass (Da)	Description	Range ^a^
1	Hex	162.0528	Hexose	0–6
2	HexNAc	203.0793	*N*-acetylhexosamine	0–5
3	Fuc	146.0579	Fucose	0–3
6	3-8NeuAc	304.1270	α2,3- or α2,8-linkage of NeuAc (SALSA: methylamine amidation)	NeuAc + NeuGc + GlcA: 0–4 GlcA: 0–1
7	6NeuAc	332.1583	α2,6-linkage of NeuAc (SALSA: isopropylamine amidation)	
8	3-8NeuGc	320.1219	α2,3- or α2,8-linkage of NeuGc (SALSA: methylamine amidation)	
9	6NeuGc	348.1532	α2,6-linkage of NeuGc (SALSA: isopropylamine amidation)	
14	GlcA	217.0950	Glucuronic acid (SALSA: isopropylamine amidation)	

**Table 2 ijms-23-13097-t002:** Updated functions of TAG.

Button Name	Function
TAG list	Generating a glycan list to be employed in TAG Expression. TAG List is a separate program written in the awk script language, as in the previous version [20]. This function is not in the menu but is executed as a separate program.
Make input file from MassList and plate file	Automatically creating TAG Expression input files from masslist and plate files.
Run TAG_Expression (calibration curve)	Glycan annotation and quantitation to generate calibration curves for annotated glycans.
Convert Excel MS files to multi-CSV files	Convert tabbed Excel files (xlsx format) to multi-CSV files for TAG Expression.

**Table 3 ijms-23-13097-t003:** New output files for the updated TAG Express. All output files are shown in Appendix A.

File Name	Description
Summary.csv	The summary file of the analysis containing the observed mass, the expression level estimated from the area, its mean value, standard deviation, CV, and the URL to search the attributed structure against Glyconnect for each plausible glycan. See main text for details.
each_glycan_quant_point.html	Data are compiled into a scatter plot for calibration curves. Linear fitting is performed, and a, b, and *R*^2^ are also outputs. Data are in HTML format and can be viewed with ordinary web browsers.
each_glycan_quant_point_rcut.html	Only glycans with *R*^2^ > 0.8 are shown in the scatter plot for the calibration curve. Linear fitting is performed, and a, b, and *R*^2^ are also outputs. Data are in HTML format and can be viewed with ordinary web browsers.

**Table 4 ijms-23-13097-t004:** Estimating glycans with *R*^2^ > 0.8 in calibration curves and hyperlink to Glyconnect database, https://glyconnect.expasy.org/browser/compositions accessed on 26 October 2022.

Theoretical Mass	Structures	Links
1362.5	(Hex)2 + (Man)3(GlcNAc)2	https://glyconnect.expasy.org/browser/compositions?f=Hex:5HexNAc:2
1403.5	(Hex)1 (HexNAc)1 + (Man)3(GlcNAc)2	https://glyconnect.expasy.org/browser/compositions?f=Hex:4HexNAc:3
1444.5	(HexNAc)2 + (Man)3(GlcNAc)2	https://glyconnect.expasy.org/browser/compositions?f=Hex:3HexNAc:4
1524.5	(Hex)3 + (Man)3(GlcNAc)2	https://glyconnect.expasy.org/browser/compositions?f=Hex:6HexNAc:2
1565.6	(Hex)2 (HexNAc)1 + (Man)3(GlcNAc)2	https://glyconnect.expasy.org/browser/compositions?f=Hex:5HexNAc:3
1590.6	(HexNAc)2 (Fuc)1 + (Man)3(GlcNAc)2	https://glyconnect.expasy.org/browser/compositions?f=Hex:3HexNAc:4dHex:1
1606.6	(Hex)1 (HexNAc)2 + (Man)3(GlcNAc)2	https://glyconnect.expasy.org/browser/compositions?f=Hex:4HexNAc:4
1647.6	(HexNAc)3 + (Man)3(GlcNAc)2	https://glyconnect.expasy.org/browser/compositions?f=Hex:3HexNAc:5
1686.6	(Hex)4 + (Man)3(GlcNAc)2	https://glyconnect.expasy.org/browser/compositions?f=Hex:7HexNAc:2
1707.6	(Hex)1 (HexNAc)1 (3-8NeuAc)1 + (Man)3(GlcNAc)2	https://glyconnect.expasy.org/browser/compositions?f=Hex:4HexNAc:3NeuAc:1
1727.6	(Hex)3 (HexNAc)1 + (Man)3(GlcNAc)2	https://glyconnect.expasy.org/browser/compositions?f=Hex:6HexNAc:3
1735.7	(Hex)1 (HexNAc)1 (6NeuAc)1 + (Man)3(GlcNAc)2	https://glyconnect.expasy.org/browser/compositions?f=Hex:4HexNAc:3NeuAc:1
1752.6	(Hex)1 (HexNAc)2 (Fuc)1 + (Man)3(GlcNAc)2	https://glyconnect.expasy.org/browser/compositions?f=Hex:4HexNAc:4dHex:1
1768.6	(Hex)2 (HexNAc)2 + (Man)3(GlcNAc)2	https://glyconnect.expasy.org/browser/compositions?f=Hex:5HexNAc:4
1793.7	(HexNAc)3 (Fuc)1 + (Man)3(GlcNAc)2	https://glyconnect.expasy.org/browser/compositions?f=Hex:3HexNAc:5dHex:1
1809.7	(Hex)1 (HexNAc)3 + (Man)3(GlcNAc)2	https://glyconnect.expasy.org/browser/compositions?f=Hex:4HexNAc:5
1848.6	(Hex)5 + (Man)3(GlcNAc)2	https://glyconnect.expasy.org/browser/compositions?f=Hex:8HexNAc:2
1869.7	(Hex)2 (HexNAc)1 (3-8NeuAc)1 + (Man)3(GlcNAc)2	https://glyconnect.expasy.org/browser/compositions?f=Hex:5HexNAc:3NeuAc:1
1897.7	(Hex)2 (HexNAc)1 (6NeuAc)1 + (Man)3(GlcNAc)2	https://glyconnect.expasy.org/browser/compositions?f=Hex:5HexNAc:3NeuAc:1
1914.7	(Hex)2 (HexNAc)2 (Fuc)1 + (Man)3(GlcNAc)2	https://glyconnect.expasy.org/browser/compositions?f=Hex:5HexNAc:4dHex:1
1938.7	(Hex)1 (HexNAc)2 (6NeuAc)1 + (Man)3(GlcNAc)2	https://glyconnect.expasy.org/browser/compositions?f=Hex:4HexNAc:4NeuAc:1
1955.7	(Hex)1 (HexNAc)3 (Fuc)1 + (Man)3(GlcNAc)2	https://glyconnect.expasy.org/browser/compositions?f=Hex:4HexNAc:5dHex:1
1971.7	(Hex)2 (HexNAc)3 + (Man)3(GlcNAc)2	https://glyconnect.expasy.org/browser/compositions?f=Hex:5HexNAc:5
2010.7	(Hex)6 + (Man)3(GlcNAc)2	https://glyconnect.expasy.org/browser/compositions?f=Hex:9HexNAc:2
	(HexNAc)3 (Fuc)1 (GlcA)1 + (Man)3(GlcNAc)2	https://glyconnect.expasy.org/browser/compositions?f=Hex:3HexNAc:5dHex:1HexA:1
2031.7	(Hex)3 (HexNAc)1 (3-8NeuAc)1 + (Man)3(GlcNAc)2	https://glyconnect.expasy.org/browser/compositions?f=Hex:6HexNAc:3NeuAc:1
2059.8	(Hex)3 (HexNAc)1 (6NeuAc)1 + (Man)3(GlcNAc)2	https://glyconnect.expasy.org/browser/compositions?f=Hex:6HexNAc:3NeuAc:1
2072.8	(Hex)2 (HexNAc)2 (3-8NeuAc)1 + (Man)3(GlcNAc)2	https://glyconnect.expasy.org/browser/compositions?f=Hex:5HexNAc:4NeuAc:1
2084.8	(Hex)1 (HexNAc)2 (Fuc)1 (6NeuAc)1 + (Man)3(GlcNAc)2	https://glyconnect.expasy.org/browser/compositions?f=Hex:4HexNAc:4dHex:1NeuAc:1
2100.8	(Hex)2 (HexNAc)2 (6NeuAc)1 + (Man)3(GlcNAc)2	https://glyconnect.expasy.org/browser/compositions?f=Hex:5HexNAc:4NeuAc:1
2141.8	(Hex)1 (HexNAc)3 (6NeuAc)1 + (Man)3(GlcNAc)2	https://glyconnect.expasy.org/browser/compositions?f=Hex:4HexNAc:5NeuAc:1
2218.8	(Hex)2 (HexNAc)2 (Fuc)1 (3-8NeuAc)1 + (Man)3(GlcNAc)2	https://glyconnect.expasy.org/browser/compositions?f=Hex:5HexNAc:4dHex:1NeuAc:1
2246.9	(Hex)2 (HexNAc)2 (Fuc)1 (6NeuAc)1 + (Man)3(GlcNAc)2	https://glyconnect.expasy.org/browser/compositions?f=Hex:5HexNAc:4dHex:1NeuAc:1
2247.8	(Hex)1 (HexNAc)3 (Fuc)3 + (Man)3(GlcNAc)2	https://glyconnect.expasy.org/browser/compositions?f=Hex:4HexNAc:5dHex:3
2287.9	(Hex)1 (HexNAc)3 (Fuc)1 (6NeuAc)1 + (Man)3(GlcNAc)2	https://glyconnect.expasy.org/browser/compositions?f=Hex:4HexNAc:5dHex:1NeuAc:1
2303.9	(Hex)2 (HexNAc)3 (6NeuAc)1 + (Man)3(GlcNAc)2	https://glyconnect.expasy.org/browser/compositions?f=Hex:5HexNAc:5NeuAc:1
2317.9	(Hex)2 (HexNAc)2 (6NeuAc)1 (GlcA)1 + (Man)3(GlcNAc)2	https://glyconnect.expasy.org/browser/compositions?f=Hex:5HexNAc:4NeuAc:1HexA:1
2327.8	(Hex)4 (HexNAc)1 (Fuc)3 + (Man)3(GlcNAc)2	https://glyconnect.expasy.org/browser/compositions?f=Hex:7HexNAc:3dHex:3
2339.9	(Hex)4 (HexNAc)1 (Fuc)1 (3-8NeuAc)1 + (Man)3(GlcNAc)2	https://glyconnect.expasy.org/browser/compositions?f=Hex:7HexNAc:3dHex:1NeuAc:1
2376.9	(Hex)2 (HexNAc)2 (3-8NeuAc)2 + (Man)3(GlcNAc)2	https://glyconnect.expasy.org/browser/compositions?f=Hex:5HexNAc:4NeuAc:2
2383.9	(Hex)5 (HexNAc)1 (6NeuAc)1 + (Man)3(GlcNAc)2	https://glyconnect.expasy.org/browser/compositions?f=Hex:8HexNAc:3NeuAc:1
2404.9	(Hex)2 (HexNAc)2 (3-8NeuAc)1 (6NeuAc)1 + (Man)3(GlcNAc)2	https://glyconnect.expasy.org/browser/compositions?f=Hex:5HexNAc:4NeuAc:2
2414.9	(Hex)5 (HexNAc)1 (Fuc)1 (GlcA)1 + (Man)3(GlcNAc)2	https://glyconnect.expasy.org/browser/compositions?f=Hex:8HexNAc:3dHex:1HexA:1
2432.9	(Hex)1 (HexNAc)5 (GlcA)1 + (Man)3(GlcNAc)2	https://glyconnect.expasy.org/browser/compositions?f=Hex:4HexNAc:7HexA:1
	(Hex)2 (HexNAc)2 (6NeuAc)2 + (Man)3(GlcNAc)2	https://glyconnect.expasy.org/browser/compositions?f=Hex:5HexNAc:4NeuAc:2
2439.9	(Hex)3 (HexNAc)2 (Fuc)2 (GlcA)1 + (Man)3(GlcNAc)2	https://glyconnect.expasy.org/browser/compositions?f=Hex:6HexNAc:4dHex:2HexA:1
2465.9	(Hex)3 (HexNAc)3 (6NeuAc)1 + (Man)3(GlcNAc)2	https://glyconnect.expasy.org/browser/compositions?f=Hex:6HexNAc:5NeuAc:1
2475.0	(HexNAc)4 (Fuc)2 (6NeuAc)1 + (Man)3(GlcNAc)2	https://glyconnect.expasy.org/browser/compositions?f=Hex:3HexNAc:6dHex:2NeuAc:1
2491.0	(Hex)1 (HexNAc)4 (Fuc)1 (6NeuAc)1 + (Man)3(GlcNAc)2	https://glyconnect.expasy.org/browser/compositions?f=Hex:4HexNAc:6dHex:1NeuAc:1
2523.0	(Hex)2 (HexNAc)2 (Fuc)1 (3-8NeuAc)2 + (Man)3(GlcNAc)2	https://glyconnect.expasy.org/browser/compositions?f=Hex:5HexNAc:4dHex:1NeuAc:2
2551.0	(Hex)2 (HexNAc)2 (Fuc)1 (3-8NeuAc)1 (6NeuAc)1 + (Man)3(GlcNAc)2	https://glyconnect.expasy.org/browser/compositions?f=Hex:5HexNAc:4dHex:1NeuAc:2
2567.0	(Hex)3 (HexNAc)2 (3-8NeuAc)1 (6NeuAc)1 + (Man)3(GlcNAc)2	https://glyconnect.expasy.org/browser/compositions?f=Hex:6HexNAc:4NeuAc:2
2579.0	(Hex)1 (HexNAc)5 (Fuc)1 (GlcA)1 + (Man)3(GlcNAc)2	https://glyconnect.expasy.org/browser/compositions?f=Hex:4HexNAc:7dHex:1HexA:1
	(Hex)2 (HexNAc)2 (Fuc)1 (6NeuAc)2 + (Man)3(GlcNAc)2	https://glyconnect.expasy.org/browser/compositions?f=Hex:5HexNAc:4dHex:1NeuAc:2
2612.0	(Hex)3 (HexNAc)3 (Fuc)1 (6NeuAc)1 + (Man)3(GlcNAc)2	https://glyconnect.expasy.org/browser/compositions?f=Hex:6HexNAc:5dHex:1NeuAc:1
2636.0	(Hex)2 (HexNAc)3 (6NeuAc)2 + (Man)3(GlcNAc)2	https://glyconnect.expasy.org/browser/compositions?f=Hex:5HexNAc:5NeuAc:2
2677.1	(Hex)1 (HexNAc)4 (6NeuAc)2 + (Man)3(GlcNAc)2	https://glyconnect.expasy.org/browser/compositions?f=Hex:4HexNAc:6NeuAc:2
2693.0	(Hex)5 (HexNAc)2 (Fuc)3 + (Man)3(GlcNAc)2	https://glyconnect.expasy.org/browser/compositions?f=Hex:8HexNAc:4dHex:3
2742.0	(Hex)3 (HexNAc)3 (3-8NeuAc)2 + (Man)3(GlcNAc)2	https://glyconnect.expasy.org/browser/compositions?f=Hex:6HexNAc:5NeuAc:2
	(Hex)2 (HexNAc)3 (Fuc)3 (6NeuAc)1 + (Man)3(GlcNAc)2	https://glyconnect.expasy.org/browser/compositions?f=Hex:5HexNAc:5dHex:3NeuAc:1
2754.1	(Hex)2 (HexNAc)3 (Fuc)1 (3-8NeuAc)1 (6NeuAc)1 + (Man)3(GlcNAc)2	https://glyconnect.expasy.org/browser/compositions?f=Hex:5HexNAc:5dHex:1NeuAc:2
2770.1	(Hex)3 (HexNAc)3 (3-8NeuAc)1 (6NeuAc)1 + (Man)3(GlcNAc)2	https://glyconnect.expasy.org/browser/compositions?f=Hex:6HexNAc:5NeuAc:2
2782.1	(Hex)2 (HexNAc)3 (Fuc)1 (6NeuAc)2 + (Man)3(GlcNAc)2	https://glyconnect.expasy.org/browser/compositions?f=Hex:5HexNAc:5dHex:1NeuAc:2
2798.1	(Hex)3 (HexNAc)3 (6NeuAc)2 + (Man)3(GlcNAc)2	https://glyconnect.expasy.org/browser/compositions?f=Hex:6HexNAc:5NeuAc:2
2831.1	(Hex)4 (HexNAc)4 (6NeuAc)1 + (Man)3(GlcNAc)2	https://glyconnect.expasy.org/browser/compositions?f=Hex:7HexNAc:6NeuAc:1
	(Hex)3 (HexNAc)2 (Fuc)2 (3-8NeuAc)2 + (Man)3(GlcNAc)2	https://glyconnect.expasy.org/browser/compositions?f=Hex:6HexNAc:4dHex:2NeuAc:2
2896.1	(Hex)5 (HexNAc)3 (Fuc)3 + (Man)3(GlcNAc)2	https://glyconnect.expasy.org/browser/compositions?f=Hex:8HexNAc:5dHex:3
2899.1	(Hex)2 (HexNAc)5 (3-8NeuAc)1 (GlcA)1 + (Man)3(GlcNAc)2	https://glyconnect.expasy.org/browser/compositions?f=Hex:5HexNAc:7NeuAc:1HexA:1
2916.1	(Hex)3 (HexNAc)3 (Fuc)1 (3-8NeuAc)1 (6NeuAc)1 + (Man)3(GlcNAc)2	https://glyconnect.expasy.org/browser/compositions?f=Hex:6HexNAc:5dHex:1NeuAc:2
2997.1	(Hex)5 (HexNAc)2 (Fuc)3 (3-8NeuAc)1 + (Man)3(GlcNAc)2	https://glyconnect.expasy.org/browser/compositions?f=Hex:8HexNAc:4dHex:3NeuAc:1
3045.2	(Hex)2 (HexNAc)5 (Fuc)1 (3-8NeuAc)1 (GlcA)1 + (Man)3(GlcNAc)2	https://glyconnect.expasy.org/browser/compositions?f=Hex:5HexNAc:7dHex:1NeuAc:1HexA:1
3074.2	(Hex)3 (HexNAc)3 (3-8NeuAc)2 (6NeuAc)1 + (Man)3(GlcNAc)2	https://glyconnect.expasy.org/browser/compositions?f=Hex:6HexNAc:5NeuAc:3
	(Hex)2 (HexNAc)3 (Fuc)3 (6NeuAc)2 + (Man)3(GlcNAc)2	https://glyconnect.expasy.org/browser/compositions?f=Hex:5HexNAc:5dHex:3NeuAc:2
3102.2	(Hex)3 (HexNAc)3 (3-8NeuAc)1 (6NeuAc)2 + (Man)3(GlcNAc)2	https://glyconnect.expasy.org/browser/compositions?f=Hex:6HexNAc:5NeuAc:3
3130.2	(Hex)3 (HexNAc)3 (6NeuAc)3 + (Man)3(GlcNAc)2	https://glyconnect.expasy.org/browser/compositions?f=Hex:6HexNAc:5NeuAc:3
3135.2	(Hex)4 (HexNAc)4 (3-8NeuAc)1 (6NeuAc)1 + (Man)3(GlcNAc)2	https://glyconnect.expasy.org/browser/compositions?f=Hex:7HexNAc:6NeuAc:2
3192.2	(Hex)3 (HexNAc)3 (Fuc)1 (3-8NeuAc)3 + (Man)3(GlcNAc)2	https://glyconnect.expasy.org/browser/compositions?f=Hex:6HexNAc:5dHex:1NeuAc:3
3248.3	(Hex)3 (HexNAc)3 (Fuc)1 (3-8NeuAc)1 (6NeuAc)2 + (Man)3(GlcNAc)2	https://glyconnect.expasy.org/browser/compositions?f=Hex:6HexNAc:5dHex:1NeuAc:3

**Table 5 ijms-23-13097-t005:** Estimating glycan composition with *R*^2^ > 0.8 of mannose-rich hybrid and LacDiNAc moieties.

Mass	Composition
Mannose-Rich Hybrid ^a^	
2383.8768	(Hex)5 (HexNAc)1 (6NeuAc)1 + (Man)3(GlcNAc)2
2414.8714	(Hex)5 (HexNAc)1 (Fuc)1 (GlcA)1 + (Man)3(GlcNAc)2
LacDiNAc	
2474.9665	(HexNAc)4 (Fuc)2 (6NeuAc)1 + (Man)3(GlcNAc)2
2677.0618	(Hex)1 (HexNAc)4 (6NeuAc)2 + (Man)3(GlcNAc)2

^a^ The hybrid structures listed here are those that are elongated to sialic acid or GlcA.

## Data Availability

The masslist and plate file, etc. for serum glycomics presented in this study will be openly available on Github https://github.com/nmiura3 under repository name IJMS2022 accessed on 26 October 2022.

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
