# Peer review of "Toolbox Accelerating Glycomics (TAG): Improving Large-Scale Serum Glycomics and Refinement to Identify SALSA-Modified and Rare Glycans"

_ijms, 2022, doi:10.3390/ijms232113097_

Round 1

Reviewer 1 Report

This manuscript is a follow-up study by the same group of authors who developed the original Toolbox Accelerating Glycomics (TAG) work in Biomolecules, 2020. In this manuscript, the authors have aimed to improve TAG for analysis of specifically sialic acid linkage-specific alkylamidation (SALSA) and Lacdinac structures that were not considered in the previous work. Although this work forms a natural next step to the previous work, it is rather poorly written with elementary figures/tables and several points that need to be addressed before the manuscript can be considered for publication.

Abstract: The abstract needs to comprehensively state the selling point of TAG in terms of other existing tools for glycomics analysis, right now it only states missing information that has been upgraded which isn’t enough – L23, what do you mean by flexible analysis? L31 – what performed well and compared to what?

Introduction: “Glycans such as glycoproteins and glycolipids” – glycoproteins and glycolipids are not types of glycans so this sentence doesn’t make sense, rephrase to something similar like “glycans decorated on proteins and lipids…”.

References 11-15 that discuss the existing software for MS-based glycomics data analysis barely includes the more prevalent software such as Skyline, Glytoucan, GlycReSoft etc. Please include those.

L58 “subglycomics other than N-glycans” – what do you mean by subglycomics?

Does TAG only work with MALDI-TOF data? Please mention this in introduction or conclusion, wherever appropriate.

How are the relative abundances of observed glycans determined using TAG?

Can the TAG list and TAG expression system be altered by the users or is it pre-built? With the constant advancements in workflows and MS instruments and acquisitions, there will be new things that will be needed to address for the regular use of this tool for data analysis – what about phosphates or sulfates or acetyl groups? Please address this in the introduction or consider adding a limitation paragraph to address what is lacking and how that can be addressed for future uses.

Experimental section: Section 3.2 – Please add more details on the SALSA method since that forms an important feature of this study. Also, add which matrix was used, and acquisition details for MALDI experiments? Section 3.3.1 is written in a very confusing manner and figure 8 doesn’t add anything to the text. Please consider rephrasing this section – maybe bring the Figure 3 to the experimental section to explain the steps better.

Results and discussion / figures and tables: Section 2.1.1. is written in a very confusing manner. I would suggest adding more information in Table 1 instead of adding in the text. In Table 1 HexNAc should be N-acetylhexosamine – please correct it.

L153 – “The TAG list has been refined….” – none of these compositions are real N-glycans – why are they considered as candidates?

Is MS/MS oxonium ions considered at any point in this toolbox?

Consider using the latest SFNG for Figure 1. Since the focus of this study is on linkage-specific sialic acids, please consider showing that in Figure 1 instead of generalized glycan classes. Please also add fucose in these cartoons. The figure doesn’t show any mannose-rich hybrids?

Table 2: By range, you mean the number of residues correct? If that is so, why is Hex only limited to 6? High mannose structures can have more than 6 Hex? You can easily combine Tables 1 and 2 and shorten the number of graphic elements.

Figure 3: Please increase font size of the smaller panels – some of them are very difficult to read.

Figure 4 is untidy. It is already an excel table where the headings can be clearly labelled instead of adding separately with arrows.

What is the point of figure 5? Does it really need to be a separate figure?

How are the lacdinac structures resolved from MALDI-TOF data?

Table 6: What glycan is Hex6? Is it missing a HexNac2? Is this data generated for the sera samples?

Other minor points:

N-glycans or N-acetylglucosamine should all be in italics

Change high-mannose to oligomannose

α symbol is messed up in many places throughout the manuscript.

Several spelling and spacing errors, duplicated words, inconsistent font sizes, throughout the manuscript – please check thoroughly.

Reference list needs to be checked, inconsistent formatting.

Reviewer 2 Report

The manuscript titled “Toolbox Accelerating Glycomics (TAG): Improving large scale serum glycomics and refinement to identify SALSA-modified and rare glycans” by Miura et al. describes improvements on TAG, a software, that was developed previously by the same authors. These improvements include adding SALSA modified linkage isomers of sialic acids to the glycan list, enabling the creation of calibration curves for the purposes of quantification, all these to enable a large-scale cohort study. The improvements are impressive and useful however I am not sure that these warrant a full paper, to me the authors should have written this up as a short communication paper - something for the authors to think about.

I have some specific comments (see below), once addressed I would recommend this manuscript for publication.

Lastly, taking a more big picture view, I would like the authors to comment on how instrument specific is TAG? I ask because the authors have used MALDI for this study and TAG seems to be designed to accommodate MALDI data. If the idea to enable more mass spectrometry users to benefit from TAG, I think that this is critical point.

 Specific comments:

1)      In certain places, authors have mentioned the words “in the previous version” to allude to the older version of TAG, e.g., page 4, line 144, I suggest that in such places, they cite their work.

2)      Page 4, lines 149- 150: Recent observations show that the spectra cannot be identified unless N-acetylgalactosamine (GalNAc) is transferred instead of Gal as shown in Figure 1c, please cite the source of the observations for readers to be able to follow.

3)      I am curious to know why the authors selected O-benzylhydroxylamine (BOA) as the label.

4)      On page 8, authors have mentioned that Neu5Gc was removed from the human serum glycan list, please explain the rationale for this as all readers may not be aware that Neu5Gc is not found in human serum.

5)      Authors should explain all numbers, and labels used in figures, for e.g., in Figure 6, please add a legend to explain the labels on each subset figure.

6) Similarly in the supporting information table, S3, the authors should define the abbreviations used like PM, PM_F etc.

Reviewer 3 Report

Manuscript ID: ijms-1932346

Title: Toolbox Accelerating Glycomics (TAG): Improving large-scale serum glycomics and refinement to identify SALSA-modified and rare glycans

Glycans, the important modification of proteins and lipids, are involved in many fundamental cellular processes. Analyzing the distribution and structure of glycans helps understand the mechanism of their function. However, it still be challenging to profile the detailed structure of glycans in bio-samples, since their complex branches, linkages and isomerization.

The author and their group contribute a lot in this field and developed powerful tools to make it easier to figure out candidate structures of glycans from MS data. Based on their previous work about Toolbox Accelerating Glycomics (TAG), this manuscript expanded the search list to cover chemical modifications of glycans, helping to determine specific linkages of terminal sialic acid sugar. It may a little evolution to the software/search engine, but a benefit to analysis the sialyl N-glycans in large-scale analysis.

The manuscript is well organized and should be published at IJMS, but not before some minor adaptations.

Suggestions:

1.    Please check the typo errors carefully. (Line 112, “current version of TAG is show shown in Table S1”; Line 131 “Glucuronic acids (GlcA) were also observed as a residue of minor acidic N-glycans”)

2.    It would be better if the author replaces the Figure 1 with high-resolution file which contains no error markers (underline).

3.    In Line 178, the word “Figure” should not be italic.

4.    Please avoid cursor mark when capturing the code of TAG List script in Figure 2.

5.    Please check the italic style of some letter, includes (but is not limited to) Line131,139,169,294,351,352,511. “N-glycans” should be “N-glycans”.

Round 2

Reviewer 1 Report

There are not further comments.